# Vascular Graft Impregnation with a Fosfomycin/Oritavancin Combination to Prevent Early Infection

**DOI:** 10.3390/pharmaceutics16111348

**Published:** 2024-10-22

**Authors:** Inês Cruz, Stefano Di Bella, Mario D’Oria, Cristina Lagatolla, M. Cristina L. Martins, Cláudia Monteiro

**Affiliations:** 1I3S—Instituto de Investigação e Inovação em Saúde, Universidade do Porto, Rua Alfredo Allen, 208, 4200-135 Porto, Portugal; inesiocruz8@gmail.com (I.C.); claudia.monteiro@i3s.up.pt (C.M.); 2INEB—Instituto de Engenharia Biomédica, Rua Alfredo Allen, 208, 4200-135 Porto, Portugal; 3Department of Medical, Surgical, and Health Sciences, University of Trieste, 34149 Trieste, Italy; stefano932@gmail.com (S.D.B.); mario.doria88@outlook.com (M.D.); 4Department of Life Sciences, University of Trieste, 34127 Trieste, Italy; clagatolla@units.it; 5ICBAS—Instituto de Ciências Biomédicas Abel Salazar, Universidade do Porto, 4050-313 Porto, Portugal

**Keywords:** vascular graft, graft infection, vascular surgery, bioengineering, antibiotics

## Abstract

**Background/Objectives:** Vascular graft infections (VGIs) represent a life-threatening complication, occurring in 0.2–6% of patients following aortic prosthetic placements. Historically, the primary focus for reducing VGIs has been on prevention. Currently, antimicrobial grafts are not available on the market. This study aimed to evaluate the efficacy of combining two antibiotics, fosfomycin and oritavancin, impregnated into the commercially available Gelweave^TM^ vascular graft as a prophylactic alternative against the most commonly implicated bacteria responsible for VGI. **Methods:** The antimicrobial activity of fosfomycin and oritavancin was assessed using the broth microdilution method, and a synergistic effect was demonstrated using the checkerboard assay against *Staphylococcus epidermidis*, methicillin-resistant *Staphylococcus aureus*, and vancomycin-resistant *Enterococcus faecium*. The antibiotics were impregnated into the commercial vascular graft through immersion, and the antimicrobial efficacy of the fosfomycin/oritavancin-impregnated graft was assessed over a period of 7 days. **Results:** Eradication of all microorganisms tested was achieved using impregnation solutions with concentrations of 40 mg/mL of fosfomycin and 256 µg/mL of oritavancin. **Conclusions:** Impregnation with the combination of fosfomycin/oritavancin proved to be a promising approach to prevent VGIs. Vascular grafts with impregnated antibiotics are not yet available on the market, and this work represents an important step toward the development of a new class of antimicrobial vascular grafts.

## 1. Introduction

Native vessels are typically preferred for vascular replacements, but in cases in which they are unavailable or unsuitable due to size or length disparities, synthetic vascular grafts (VGs) are utilized. While VG implantation has high technical and clinical success rates, implant failure may occur in a significant number of patients due to several reasons, the most serious of which is vascular graft infection (VGI), which occurs in 0.2–6% of patients. [1]. Indeed, this condition is a life-threatening complication and remains one of the most dreaded problems associated with vascular graft reconstructive surgery [2]. Infections caused by Gram-positive bacteria, including methicillin-susceptible *Staphylococcus aureus* (MSSA), methicillin-resistant *Staphylococcus aureus* (MRSA), and coagulase-negative staphylococci, such as *Staphylococcus epidermidis*, accounts for more than 50% of vascular graft infections [3]. In addition, vancomycin-resistant *Enterococcus* (VRE) is a challenging and emerging pathogen responsible for prosthetic infections [4]. Contamination typically occurs during implantation, either from microorganisms colonizing the patient’s skin or contaminants from surgical materials [5]. However, infections can also arise later due to dissemination from distant foci or owing to direct colonization from nearby sources, such as aorto-esophageal and aorto-enteric fistulas [2,6].

Primary strategies to reduce VGIs have focused on prevention, including the development of antimicrobial vascular grafts [7]. In vitro and in vivo studies have demonstrated that polyester materials sealed with collagen or gelatin showed greater efficacy due to reduced blood loss, faster healing, and increased endothelialization [8]. Bonding of antibiotics to gelatin-coated knitted Dacron^®^ vascular grafts confers antimicrobial properties to the material for several days and minimizes the possibility of VGI by delivering antibiotics at the time when the graft is at the greatest risk of contamination. In fact, rifampicin bonding to protein-sealed VGs is easily obtained and widely used in emergency situations by soaking commercially available polyester vascular grafts in a 60 mg/mL solution of rifampicin for 15 min [6,9]. Nevertheless, rifampicin-impregnated VGs have proven ineffective over time due to the emergence of resistant bacteria to this antibiotic and showed reinfection rates of up to 22% after replacement [3,6,10]; moreover, they are not currently available on the market. Recent studies have demonstrated that the InterGard^®^ Synergy (IGSys) polyester vascular graft containing a combination of silver and triclosan exhibited greater in vitro antimicrobial activity compared to a vascular graft containing silver alone [3]. However, silver grafts may induce bacterial resistance, and toxic effects have been reported due to systemic absorption of silver ions in chronic exposure [11].

This work aims to overcome the limitations faced by the current antimicrobial vascular grafts by impregnating a combination of fosfomycin and oritavancin on a commercial synthetic vascular graft. Fosfomycin is a well-known antibiotic with a low molecular weight and a broad spectrum of activity against both Gram-negative and Gram-positive pathogens, specifically staphylococci [12]. This antibiotic inhibits the enzyme UDP-N-acetylglucosamine enolpyruvyl transferase (MurA), which is responsible for catalyzing the formation of N-acetylmuramic acid, which is necessary for peptidoglycan synthesis [13,14]. Fosfomycin also has good anti-biofilm properties and, when used in combination with other classes of antibiotics, provides a synergistic effect, significantly increasing bacterial killing [12,15,16]. Oritavancin is a new semisynthetic lipoglycopeptide analog of vancomycin, with a long-acting effect [17]. This antibiotic has a hydrophobic side chain that facilitates interaction with the bacterial cell membrane and inhibits transpeptidation and transglycosylation, leading to the disruption of the membrane ultrastructure [18]. Oritavancin revealed a high in vitro antimicrobial activity against both resistant and susceptible Gram-positive organisms, including *Staphylococcus*, namely MRSA, *Streptococcus*, and *Enterococcus* [19]. Additionally, oritavancin showed a synergistic effect when combined with fosfomycin against VRE isolates [16,20]. Synergism between fosfomycin and oritavancin can be a potential solution for the prevention of VGIs, as this combination could be useful to prevent resistance, increase bacterial killing, and prevent bacterial adhesion and biofilm formation.

## 2. Materials and Methods

### 2.1. Antibiotic Stock Solution Preparation

Stock solutions for each antibiotic were prepared according to the Clinical and Laboratory Standards Institute guidelines [21].

A 2560 μg/mL fosfomycin (P5396; Merck; Darmstadt, Germany) stock solution was prepared in distilled water and stored at 4 °C. Oritavancin (FO58233; Biosynth; Compton, UK) stock solution was prepared at a concentration of 640 μg/mL in distilled water with 0.002% of polysorbate 80 (Tween^TM^ 80; Fisher BioReagents; Waltham, MA, USA) and stored at −20 °C.

### 2.2. Bacterial Strains and Culture Conditions

Two reference strains were used, *Staphylococcus epidermidis* (ATCC 35984) and MRSA (ATCC 33591), as well as a VanA *Enterococcus faecium* (VREf-10) clinical isolate, for which the combination of fosfomycin plus oritavancin was shown to be synergistic in a previous study [15].

All strains, which were stored at −80 °C in cryovials, were spread on Tryptic Soy Agar plates (TSA; Merck; Darmstadt, Germany) and incubated overnight at 37 °C. Bacterial inoculum was prepared by resuspending 1–2 colonies of bacteria in 5 mL of Mueller–Hinton Broth (MHB; Merck; Darmstadt, Germany) and incubated overnight at 37 °C, 150 rpm.

### 2.3. Minimum Inhibitory Concentration (MIC) and Minimum Bactericidal Concentration (MBC) Assays

The antimicrobial activity of fosfomycin and oritavancin against *S. epidermidis*, MRSA, and VREf-10 was performed using the broth microdilution method in a 96-well polypropylene microtiter plate (3879; Corning^®^; Kennebunk, ME, USA) [22]. Twofold serial dilutions of the antibiotic were prepared, and 10 μL of each dilution was dispensed into the plate containing 90 μL of the inoculum, which was adjusted, using optical density (OD) at 600 nm, to 2 × 10^5^ colony forming units (CFUs) per mL, in MHB. To test the antimicrobial activity of fosfomycin against VREf, the MHB medium was supplemented with glucose-6-phosphate (G6P) 25 µg/mL. In addition, polysorbate 80 (Tween^TM^ 80) 0.002% was added to the MHB medium for oritavancin susceptibility testing against all strains. Then, the plate was incubated at 37 °C for 18–24 h, and bacterial growth was evaluated through the visual observation of bacteria deposition on the bottom of the plate. The minimum inhibitory concentration (MIC) of the antibiotics was defined as the lowest concentration capable of inhibiting visible growth. The contents of the first 3 wells, in which no growth was observed, were serially diluted in sterile phosphate-buffered saline (PBS) solution, plated on TSA plates, and incubated overnight at 37 °C for CFU counting. The minimum bactericidal concentration (MBC) was defined as the lowest concentration that reduces ≥ 99.9% of the initial inoculum.

### 2.4. Synergy Assays

The determination of the susceptibility of *S. epidermidis*, MRSA, and VREf-10 to fosfomycin and oritavancin in combination was performed using a checkerboard assay in a 96-well microtiter plate [23]. The oritavancin solution was twofold serially diluted along each column of the plate, and twofold serially diluted fosfomycin solution was placed along each row. A previously cultivated inoculum, as described above, was adjusted to 2 × 10^5^ CFUs/mL by OD at 600 nm in MHB with TweenTM 80 (0.002%) and placed on the 96-well plate in contact with the antibiotic dilutions. In the synergism assays against VREf, the medium was also supplemented with G6P 25 µg/mL. The MICs of the antibiotics alone were determined along the last two rows of the plate, and the negative control (MHB) and positive control (inoculum) were placed in the last two columns. After incubation at 37 °C for 18–24 h, the fractional inhibitory concentration index (FICI) of the antibiotics was calculated according to the following formulation:$$\mathrm{FICI}=\frac{\mathrm{MIC}\;A\;\mathrm{in}\;\mathrm{combination}}{\mathrm{MIC}\;A\;\mathrm{alone}}+\frac{\mathrm{MIC}\;B\;\mathrm{in}\;\mathrm{combination}}{\mathrm{MIC}\;B\;\mathrm{alone}\;},$$
where A is oritavancin, B is fosfomycin, and MIC is the minimum inhibitory concentration. Synergism occurs when the FICI ≤ 0.5. An additive effect occurs when 0.5 < FICI ≤ 1, no interaction is defined as 1 < FICI ≤ 4, and antagonism occurs when FICI > 4.0 [16].

### 2.5. Preparation of Fosfomycin/Oritavancin-Impregnated Vascular Graft

The Gelweave^TM^ (734032, Terumo Aortic; Inchinnan, UK) gelatin-coated vascular graft was cut into disks with a 6 mm diameter punch and sterilized in a laminar flow hood using ultraviolet (UV) light (365 nm) for 15 min on each side. Afterward, the samples were immersed for 30 min in 500 µL of different antibiotic solutions, as described in Table 1, and only with distilled water as a control. After incubation, the samples were dried with a stream of argon and stored in an exicator. Fosfomycin concentrations were selected based on previous studies, in which rifampicin was impregnated in similar grafts [3]. Oritavancin concentrations were selected based on the MIC results. For each strain, antibiotic concentrations were adjusted according to the efficiency obtained.

### 2.6. Antimicrobial Evaluation of Fosfomycin/Oritavancin-Impregnated Vascular Graft

The antimicrobial efficacy of the fosfomycin/oritavancin graft was performed according to Xavier Berard et al. for 7 days against *S. epidermidis*, MRSA, and VREf-10 [3]. This methodology was used as an alternative to the standard agar diffusion plate test (ISO 20645:2004 [24]) for determining the antibacterial activity of textile/fiber materials, allowing for the precise quantification of the bacterial burden to be eliminated by the grafts while simulating graft–liquid (blood) contact.

The ability of the grafts to eliminate planktonic and adherent bacteria was determined. The VG samples, which had been previously prepared, were immersed in a 1.5 mL Eppendorf tube containing 500 µL of bacterial inoculum adjusted to 10^5^ CFUs/mL in MHB by OD at 600 nm. Samples were then incubated at 37 °C, and the antimicrobial efficacy was evaluated over time. For this purpose, after 1, 2, 3, and 7 days, 100 μL of the broth culture medium was collected from each sample, diluted in PBS, plated on TSA plates, and incubated at 37 °C for 18 h. In addition, 500 μL of fresh medium was added to all samples on day 3. The absence of bacterial growth on the TSA plates would indicate that the antibiotic-impregnated VG samples were able to release the impregnated antibiotics and kill planktonic bacteria in the broth culture medium. Sonication of the antibiotic-impregnated VGs was performed to assess the number of viable adherent bacteria on the graft sample surface at day 7. For that, graft samples were removed from the bacterial liquid medium, washed 3 times with PBS, and sonicated in an ultrasonic bath (Bactosonic, Bandelin; Berlin, Germany) 200 W for 5 min in 500 µL of MHB. Afterward, graft samples were vortexed for 20 s, and the medium was serially diluted in PBS, plated on TSA plates, and incubated at 37 °C, for 18 h for CFU counting [4]. The absence of bacterial growth on the TSA plates would indicate that the antibiotic-impregnated VG samples could inhibit bacterial adhesion or induce the killing of adherent bacteria.

## 3. Results

### 3.1. MIC and MBC Assays

The susceptibility of *S. epidermidis*, MRSA, and VREf-10 to fosfomycin and oritavancin was determined by performing the MIC and MBC assays, as shown in Table 2. Fosfomycin proved to be the most effective antibiotic against *S. epidermidis*, with MIC values between 0.5 and 1 µg/mL. However, for MRSA and VREf-10, fosfomycin demonstrated lower efficacy with higher MIC values of 16–32 µg/mL for MRSA and 128 µg/mL for VREf-10. The bactericidal activity of fosfomycin against *S. epidermidis* was detected at 2 µg/mL, and no bactericidal effect was detected at the tested concentrations for MRSA and VREf-10. Oritavancin showed MIC values between 1 and 2 µg/mL for *S. epidermidis* and proved to be highly effective against MRSA and VREf-10, with MIC value of 0.5 µg/mL for both bacteria. The bactericidal activity of oritavancin was detected at the same concentrations of MIC for *S. epidermidis* (1–2 µg/mL) and between 0.5 and 1 µg/mL for MRSA. No bactericidal effect of oritavancin against VREf-10 was detected at the tested concentrations.

### 3.2. Synergy Assays

The synergistic effect of fosfomycin and oritavancin was tested using the checkerboard assay against *S. epidermidis*, MRSA, and VREf-10. FICI values for the combinations ranging from 0.157 to 0.374 for *S. epidermidis*, 0.365–0.498 for MRSA, and 0.5 for VREf-10 demonstrated synergism between the two antibiotics against all the tested strains (Table 3). The most effective combinations were 1 µg/mL of fosfomycin + 0.016 µg/mL of oritavancin against *S. epidermidis* and 1 µg/mL of fosfomycin + 0.031 µg/mL of oritavancin against MRSA. The synergistic effect against VREf-10 was only detected in one combination: 32 µg/mL of fosfomycin + 0.062 µg/mL of oritavancin.

### 3.3. Antimicrobial Evaluation of Fosfomycin/Oritavancin-Impregnated Vascular Graft

#### 3.3.1. *S. epidermidis*

After confirmation of synergism between fosfomycin and oritavancin against all the tested strains, the vascular graft was impregnated with both antibiotics. VGs impregnated with only one antibiotic, fosfomycin 0.62 mg/mL and 1.25 mg/mL and oritavancin 16 µg/mL and 32 µg/mL, allowed bacterial growth during the 7 days (Figure 1). Using the combination of fosfomycin/oritavancin, bacterial eradication from day 1 and persistent for 7 days was observed at a concentration of 1.25 mg/mL of fosfomycin combined with 16 and 32 µg/mL of oritavancin. This demonstrates that VGs impregnated with the combination of fosfomycin/oritavancin are more efficient than the grafts impregnated with only one antibiotic at the same concentration. In addition, when the fosfomycin/oritavancin VG was impregnated with a lower concentration of fosfomycin (F0.62) and the same concentrations of oritavancin (O16 and O32), bacterial growth occurred over the 7 days, suggesting that a higher concentration of impregnated fosfomycin (1.25 mg/mL) is required to achieve the antimicrobial effect. Regarding the control of VG prepared in distilled water (graft + water), results showed bacterial growth over the 7 days, as expected.

After sonication on day 7, VGs impregnated with each antibiotic at the concentrations tested (F1.25, F0.62, O16, and O32) showed live adherent bacteria on the graft sample surfaces, as expected (Figure 2). In the fosfomycin/oritavancin-impregnated VGs, no live adherent bacteria were detected at the same concentrations of the bactericidal combinations previously obtained in the incubation medium (1.25 mg/mL of fosfomycin and 16/32 μg/mL of oritavancin). This demonstrates that the impregnation of the two antibiotics into the VG is also efficient regarding the inhibition of adhesion of live *S. epidermidis* to the graft surface at these concentrations. Finally, the control of sonicated VG prepared in distilled water (graft + water) showed bacterial growth, indicating that there were live adherent bacteria on the surface of the graft samples, as expected.

#### 3.3.2. MRSA

The results of antimicrobial assays of fosfomycin/oritavancin-impregnated VGs against MRSA are presented in Figure 3. Although no bacterial eradication was detected in VGs impregnated with a single antibiotic, at 40 mg/mL of fosfomycin and 256 and 512 µg/mL of oritavancin, bacterial growth was inhibited over 7 days when compared to the control of the graft prepared in water. Regarding the antimicrobial efficacy of the fosfomycin/oritavancin VGs, although no bacterial eradication was detected over 7 days, with fosfomycin 40 mg/mL and oritavancin 64/128 µg/mL, there was bacterial growth inhibition compared to the control of MRSA culture. Bacterial eradication was achieved only after the second day at concentrations of impregnated antibiotics in a combination of 40 mg/mL fosfomycin and 256/512 μg/mL oritavancin. These results reflect the high resistance of MRSA to fosfomycin, which was previously observed in the MIC and MBC assessment shown in Table 2, in which it was not possible to obtain an MBC value for fosfomycin against MRSA for the concentrations tested. However, the increased release of fosfomycin and oritavancin over time is sufficient to achieve bacterial eradication from the second day onwards. The control performed as expected, showing bacterial growth over the 7 days in the VG prepared in distilled water (graft + water).

By day 7, the fosfomycin/oritavancin-impregnated VGs were sonicated, and the CFU counts of live adherent bacteria are shown in Figure 4. VGs impregnated with a single antibiotic at the concentrations tested (F40, O128, O256, and O512) showed live adherent bacteria on the graft sample surfaces, and VGs impregnated with the two antibiotics in combination at a concentration of 40 mg/mL fosfomycin and 64/128 µg/mL oritavancin showed live adherent bacteria, as expected considering the results of the incubation media. On the contrary, no live adherent bacteria were detected in the VGs impregnated with fosfomycin/oritavancin in the same combinations that showed a bactericidal effect in the incubation media (40 mg/mL fosfomycin and 256/512 μg/mL oritavancin). This demonstrates that the fosfomycin/oritavancin-impregnated VGs are capable of inhibiting the adhesion of live MRSA at these concentrations. Control of sonicated VG prepared in distilled water (graft + water) showed bacterial growth, indicating that there were live adherent bacteria on the surface of the graft samples, as expected.

#### 3.3.3. VREf-10

The antimicrobial activity of fosfomycin/oritavancin-impregnated VGs against VREf-10 is presented in Figure 5. Although no bacterial eradication was detected in the VGs impregnated with each antibiotic, at the concentrations of 40 mg/mL of fosfomycin, and 128 and 256 μg/mL of oritavancin, bacterial growth was reduced when compared to the controls of VREf-10 culture and the graft prepared in water, especially in the first days of incubation. Furthermore, bacterial eradication was observed in VGs impregnated with 512 μg/mL of oritavancin from day 3, demonstrating that O512 is effective against VREf-10. In addition, bacterial eradication was detected by day 7 in VGs impregnated with antibiotic combinations at concentrations of 40 mg/mL fosfomycin and 128/256/512 μg/mL oritavancin. The control VG, which was prepared in distilled water (graft + water), showed bacterial growth over the 7 days, as expected. These results demonstrate that the most efficient treatment was O512 μg/mL and that the combination of the two antibiotics against VREf-10 was only beneficial for lower concentrations of oritavancin (O128 and O256).

Regarding the live adherent bacteria on the surface of the fosfomycin/oritavancin-impregnated VGs, the CFU counts of the sonicated samples are shown in Figure 6. VGs impregnated with a single antibiotic at concentrations F40, O128, and O256 showed live adherent bacteria on the surface. Nonetheless, as expected from results in the incubation media, no live adherent bacteria were observed on the surface of the VGs impregnated with oritavancin 512 μg/mL (O512). Moreover, no live adherent bacteria were detected in the fosfomycin/oritavancin-impregnated VGs at the same concentrations of the previously obtained combinations (40 mg/mL of fosfomycin and 128/256/512 μg/mL of oritavancin). The control VG, which was prepared in distilled water (graft + water), presented live adherent bacteria on the surface of the graft samples, as expected.

## 4. Discussion

When VGIs occur, a complete graft explantation (coupled with targeted antibiotic therapy) is regarded as the treatment of choice and is recommended by clinical practice guidelines. However, mortality from VGIs, even after radical surgical explantation, remains cumbersome, as shown in several publications. Therefore, the prevention of VGIs still represents an area with potential for major improvements, and our study delineates a novel option for expanding the armamentarium of vascular devices. The antibiotic-impregnated graft described in this work may also serve as reconstruction material for infective native aortic aneurysms (also known as “mycotic aneurysms”) when the etiologic bacterium is known and corresponds to a sensible strain. Furthermore, when a VGI has occurred and a need for replacement of the infected graft arises, the matter of how to reconstruct the patient’s vessels becomes even more challenging. While the use of autologous veins is regarded as the first-line option, they may not be of adequate quality for extensive replacement, and several options (such as cryopreserved allografts or bovine pericardium) have emerged [25]. In a recent systematic review, it was found that the literature focused on direct comparison between different types of VGs is scarce, particularly when related to materials other than autologous veins [26]. Although the authors found a lower overall mortality rate in patients treated with biological material or with autologous veins only, in recent reports, prostheses have provided promising results in terms of mortality and reinfection rate. In fact, one major advantage of prosthetic grafts is their rapid on-stock availability and ease of handling, which makes them the most suitable alternative, especially in urgent circumstances. As such, the proposed graft is very promising to pursue in vivo studies to confirm efficacy in the prevention of VGIs [25,26,27,28,29]. Additional tests to assess the time and amount of antibiotics released and how long they remain impregnated in the VG would be also relevant to the study. The safety of the VG impregnated with oritavancin and fosfomycin, with regard to potential drug-related side effects, also needs to be confirmed. Although both antibiotics are already in clinical use and their biocompatibility has been extensively studied [30,31], with no common side effects reported, the potential for allergic reactions should still be considered [32,33]. In vivo tests would also allow us to assess the biodistribution of antibiotics in organs and possible side effects. Moreover, a disrupting effect on gut flora may not be excluded.

## 5. Conclusions

Overall, the combination of fosfomycin and oritavancin enhances the antimicrobial properties of the gelatin-sealed VGs against the main bacteria involved in VGIs (*S. epidermidis*, MRSA, and VREf) compared to the impregnation of each antibiotic alone, namely at concentrations of 40 mg/mL of fosfomycin and 256 µg/mL of oritavancin.

The developed strategy overcomes antibiotic resistance-associated problems, covering a broad spectrum of bacteria and inhibiting bacterial adhesion and, consequently, biofilm formation and infection establishment. In addition, the combination allows the use of lower amounts of each drug. A future application of this approach would require a pre-clinical study to evaluate efficiency and safety. Regarding translation to industry, impregnation of the antibiotics in the graft coating during production process would be interesting to explore. This is a promising new approach with strong potential for the prevention and treatment of VGIs.

## Figures and Tables

**Figure 1 pharmaceutics-16-01348-f001:**
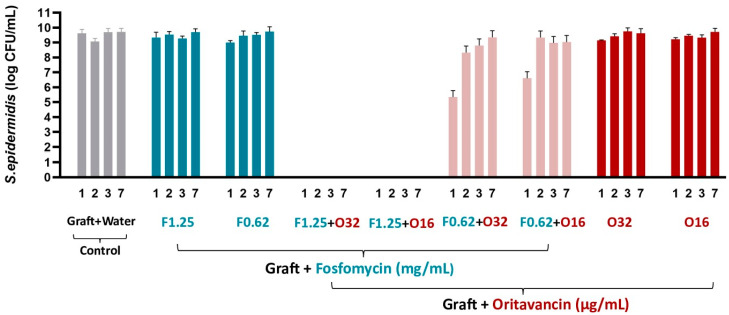
Colony forming unit (CFU) counts of *S. epidermidis* measured after incubation in media with fosfomycin (F) and/or oritavancin (O) impregnated VGs and control (graft prepared in water) on days 1, 2, 3, and 7.

**Figure 2 pharmaceutics-16-01348-f002:**
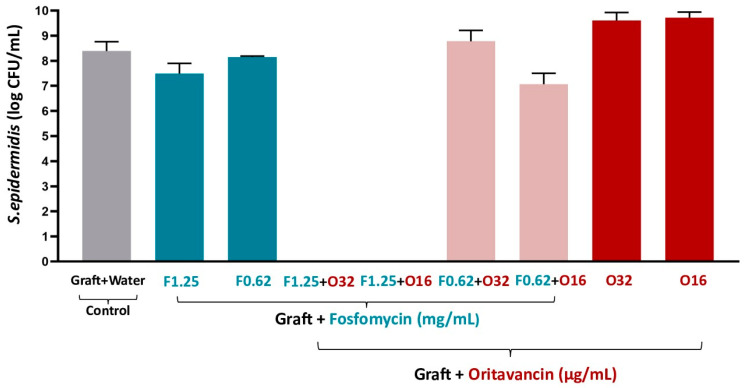
Colony forming unit (CFU) counts of sonicated fosfomycin (F)/oritavancin (O)-impregnated VGs and control (graft prepared in water) on day 7 of incubation with *S. epidermidis*.

**Figure 3 pharmaceutics-16-01348-f003:**
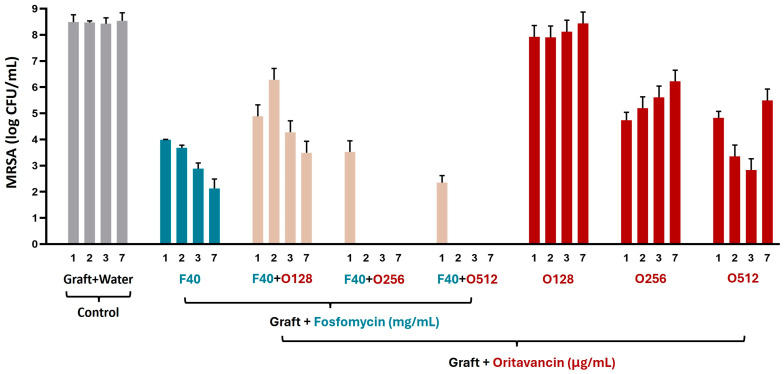
Colony forming unit (CFU) counts of methicillin-resistant *Staphylococcus aureus* (MRSA) measured after incubation in media with fosfomycin (F)- and/or oritavancin (O)-impregnated VGs and control (graft prepared in water) on days 1, 2, 3, and 7.

**Figure 4 pharmaceutics-16-01348-f004:**
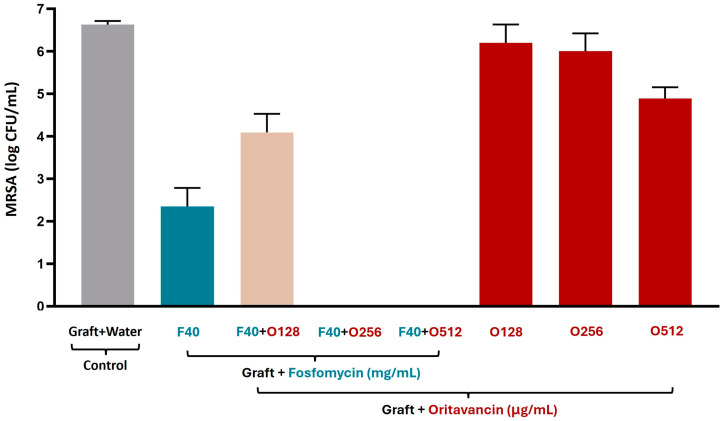
Colony forming unit (CFU) counts of sonicated fosfomycin (F)/oritavancin (O)-impregnated VGs and control (graft prepared in water) on day 7 of incubation with MRSA.

**Figure 5 pharmaceutics-16-01348-f005:**
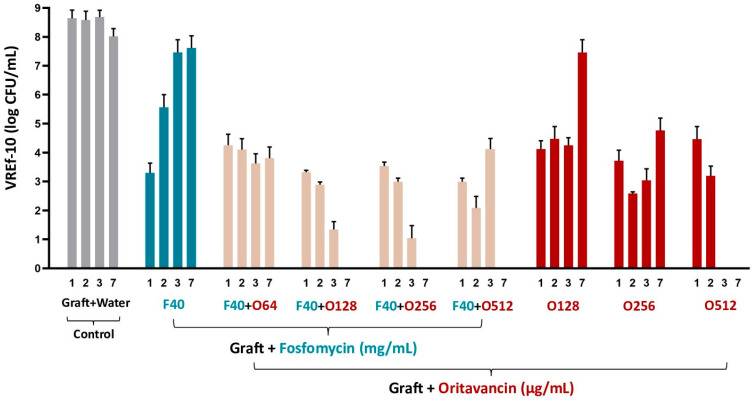
Colony forming unit (CFU) counts of vancomycin-resistant *Enterococcus faecium* (VREf-10) measured after incubation in media with fosfomycin (F)- and/or oritavancin (O)-impregnated VGs and control (graft prepared in water) on days 1, 2, 3, and 7.

**Figure 6 pharmaceutics-16-01348-f006:**
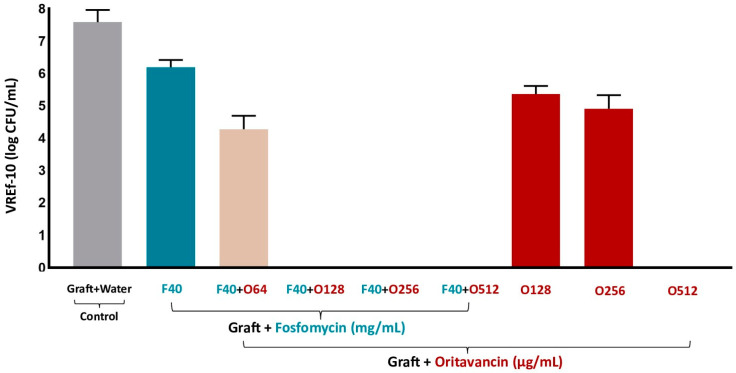
Colony forming unit (CFU) counts of sonicated fosfomycin (F)/oritavancin (O)-impregnated VGs and control (graft prepared in water) on day 7 of incubation with VREf-10.

**Table 1 pharmaceutics-16-01348-t001:** Concentrations of the antibiotic solutions used to prepare the antibiotic-impregnated vascular grafts (VG) to test its antimicrobial efficacy against *Staphylococcus epidermidis* (*S. epidermidis*), methicillin-resistant *Staphylococcus aureus* (MRSA), and VanA *Enterococcus faecium* (VREf-10).

Antibiotic Solutions	*S. epidermidis*	MRSA and VREf-10
Fosfomycin (mg/mL)	1.25/0.625	40
+	+	+
Oritavancin (μg/mL)	16/32/64	64/128/265/512

**Table 2 pharmaceutics-16-01348-t002:** Antimicrobial activity in µg/mL concerning minimum inhibitory concentration (MIC) and minimum bactericidal concentration (MBC) of fosfomycin and oritavancin against *Staphylococcus epidermidis* (*S. epidermidis*), methicillin-resistant *Staphylococcus aureus* (MRSA), and VanA *Enterococcus faecium* (VREf-10).

Antimicrobial Agent		*S. epidermidis* (ATCC 35984)	MRSA (ATCC 33591)	VREf-10 (Clinical Strin)
Fosfomycin	MIC	0.5–1	16–32	128
	MBC	2	>128	>512
Oritavancin	MIC	1–2	0.5	0.5
	MBC	1–2	0.5–1	>2

**Table 3 pharmaceutics-16-01348-t003:** Minimum inhibitory concentration (MIC) values in µg/mL of fosfomycin (F) and oritavancin (O), alone and in combination, obtained using the checkerboard assay, against *Staphylococcus epidermidis* (*S. epidermidis*), methicillin-resistant *Staphylococcus aureus* (MRSA), and VanA *Enterococcus faecium* (VREf-10) and the resulting fractional inhibitory concentration index (FICI) values < 0.5 (synergistic effect).

*S. epidermidis* (ATCC 35984)					MRSA (ATCC 33591)					VREf-10 (Clinical Strin)				
MIC Alone		MIC in Combination		FICI	MIC Alone		MIC in Combination		FICI	MIC Alone		MIC in Combination		FICI
F	O	F	O		F	O	F	O		F	O	F	O	
8	0.5	2	0.008–0.016	0.266–0.282	8	0.250	2	0.062	0.498	128	0.25	32	0.062	0.5
8		1	0.016	0.157		0.125	1–2	0.031	0.373–0.498					
4		1	0.008–0.062	0.266–0.374		0.062	2	0.008	0.379					
4		0.5	0.031	0.187										

## Data Availability

The original contributions presented in the study are included in the article, further inquiries can be directed to the corresponding author.

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
