# Peer review of "Vascular Graft Impregnation with a Fosfomycin/Oritavancin Combination to Prevent Early Infection"

_pharmaceutics, 2024, doi:10.3390/pharmaceutics16111348_

Round 1
Reviewer 1 Report
Comments and Suggestions for Authors
The manuscript presents a compelling study on the pre-clinical efficacy of a combination of fosfomycin and oritavancin impregnated into Gelweave™ vascular grafts to prevent early infections. The study addresses a significant clinical challenge of vascular graft infections (VGIs) and proposes a novel prophylactic approach. The methodology is well-defined, and the promising results show potential for future clinical applications. However, some areas require further clarification and elaboration to strengthen the overall quality of the manuscript.
Introduction:
Lines 65 - 67: This statement needs a reference.
Discussion:
Lines 335 - 341: This paragraph could benefit from elaborating on the specific side effects expected from using the combination of fosfomycin and oritavancin, citing previous research on the prevalence of those side effects. Also, references are missing from this paragraph, even though it states some facts that need references to support them.
The discussion section should include a paragraph addressing the study's potential limitations.
Conclusion:
The conclusion could be enhanced by highlighting future research directions and the steps needed to advance this approach toward clinical application.
Tables:
The abbreviations in the tables need to be spelled out in the footnotes.
Author Response
The manuscript presents a compelling study on the pre-clinical efficacy of a combination of fosfomycin and oritavancin impregnated into Gelweave™ vascular grafts to prevent early infections. The study addresses a significant clinical challenge of vascular graft infections (VGIs) and proposes a novel prophylactic approach. The methodology is well-defined, and the promising results show potential for future clinical applications. However, some areas require further clarification and elaboration to strengthen the overall quality of the manuscript.
Introduction:
Lines 65 - 67: This statement needs a reference.
Thank you for the suggestion. The following statement was carefully corrected and a reference regarding safety and efficiency of silver was added.
Lines: 63-65
Before:
“Although silver grafts do not contribute to antibiotic resistance, toxic effects have been reported due to systemic absorption of silver ions in chronic exposure.
After:
“However, silver grafts may induce bacterial resistance and toxic effects have been reported due to systemic absorption of silver ions in chronic exposure [11].
REF: Cutting K, White R, Edmonds M. The safety and efficacy of dressings with silver - addressing clinical concerns. Int Wound J. 2007 Jun;4(2):177-84.
Discussion:
Lines 335 - 341: This paragraph could benefit from elaborating on the specific side effects expected from using the combination of fosfomycin and oritavancin, citing previous research on the prevalence of those side effects. Also, references are missing from this paragraph, even though it states some facts that need references to support them. The discussion section should include a paragraph addressing the study's potential limitations.
Thank you for the suggestion. Oritavancin and fosfomycin are generally considered safe antibiotics. However, although rare, there is a potential for allergic reactions for each antibiotic when used alone, which may also happen when used in combination.
For clarification, we have included this information and new references at the end of the discussion:
“The safety of the VG impregnated with oritavancin and fosfomycin related to potential drug-related side effects also needs to be confirmed. Although there are not common side effects of oritavancin and fosfomycin reported, the potential for allergic reactions should be considered [29, 30].”
Conclusion:
The conclusion could be enhanced by highlighting future research directions and the steps needed to advance this approach toward clinical application.
Future research directions were added to the conclusions:
“A future application of this approach would require a pre-clinical study to evaluate efficiency and safety. Regarding translation to industry, impregnation of the antibiotics in the grafts during production process would be interesting to explore.”
Tables:
The abbreviations in the tables need to be spelled out in the footnotes.
The abbreviations in the tables were spelled in the footnotes.

Reviewer 2 Report
Comments and Suggestions for Authors
The manuscript "Vascular grafts impregnation with fosfomycin/oritavancin combination to prevent early infection" by I. Cruz et al. is devoted to the development of antibiotic-impregnated vascular grafts. The authors described optimization of antibiotics use due to synergistic action for 3 tested gram-positive bacterial strains. As well as the samples of impregnated material were tested by incubation with bacterial suspension. The results are not so clear for the reviewer due to several main points:
- So, vascular grafts is a highly competitive area of research and development. There is a standard approach for testing of antibiotic-treated textile/fiber materials (ISO 20645:2004. Textile Fabrics - Determination of antimicrobial activity - Agar diffusion plate test). Why the authors did not use this simple qualitative approach?
- What about cytotoxicity of the prepared materials sample? Oritavancin is a highly lipophilic compound recommended for topical use not systemic.
- Bacterial test strains is represented only by gram-positive bacteria (staphylococci and enterococci). Why the authors did not use relevant gram-negative bacteria (E. coli etc.)?
- The E. faecium isolate should be properly characterized. At least, the type of Van resistance should be disclosed.
Without any clarification of the listed details the presented work seems incomplete and speculative. It could not be accepted for publication in the current form.
Author Response
The manuscript "Vascular grafts impregnation with fosfomycin/oritavancin combination to prevent early infection" by I. Cruz et al. is devoted to the development of antibiotic-impregnated vascular grafts. The authors described optimization of antibiotics use due to synergistic action for 3 tested gram-positive bacterial strains. As well as the samples of impregnated material were tested by incubation with bacterial suspension. The results are not so clear for the reviewer due to several main points:
- So, vascular grafts is a highly competitive area of research and development. There is a standard approach for testing of antibiotic-treated textile/fiber materials (ISO 20645:2004. Textile Fabrics - Determination of antimicrobial activity - Agar diffusion plate test). Why the authors did not use this simple qualitative approach?
Thank you for the question. Although we were aware of the ISO for textile fabrics we believe that the test we performed is more suitable for the application of the grafts. The agar diffusion plate test depends on the ability of the antibiotics to diffuse in the agar and a precise quantification of the bacterial burden that the grafts are able to eliminate is not possible to obtain using this method. The test we applied mimics the contact of the graft with blood, a liquid-like contact and allowed precise results regarding the number of eliminated bacteria.
- What about cytotoxicity of the prepared materials sample? Oritavancin is a highly lipophilic compound recommended for topical use not systemic.
Oritavancin is not recommended for topical use. It is an antibiotic administered intravenously. It is used to treat both localized and systemic infections.
- Bacterial test strains is represented only by gram-positive bacteria (staphylococci and enterococci). Why the authors did not use relevant gram-negative bacteria (E. coli etc.)?
Gram-positive strains are much more relevant and predominant in this kind of infection than Gram-negative strains, as described in the sentence (lines 38-41) of the introduction:
“Infections caused by Gram-positive bacteria, including methicillin-susceptible Staphylococcus aureus (MSSA), methicillin-resistant Staphylococcus aureus (MRSA) and coagulase-negative staphylococci such as Staphylococcus epidermidis, accounts for more than 50% of vascular graft infections [3]”
Therefore, this study focused on the most relevant Gram-positive bacteria.
- The E. faecium isolate should be properly characterized. At least, the type of Van resistance should be disclosed.
The VanA determinant of the strain and a reference to an earlier study in which it was used have been added to the material and methods section of the manuscript.
“Two reference strains were used, Staphylococcus epidermidis (ATCC 35984) and MRSA (ATCC 33591), as well as a VanA Enterococcus faecium (VREf-10) clinical isolate, for which the combination of fosfomycin plus oritavancin was shown to be synergistic in a previous study [15].”

Reviewer 3 Report
Comments and Suggestions for Authors
In general, I have an average opinion of the article. Apparently, they were inspired by their previous work [15], where a synergistic effect was detected when fosfomycin was combined with oritavancin. Therefore, they decided to conduct this study. The authors have examined how the combination of fosfomycin and oritavancin can suppress the growth of gram-positive bacteria on the surface of vascular grafts.
I have some comments that I believe should be considered.
Line 96: Bacterial strains... (not one strain).
The MIC data for individual drugs in Table 3 does not match the values specified in Table 2. Additionally, it is unclear which MIC values were used for the FIC calculations, as the MIC values are given in a range. Please provide an explanation.
Lines 210-213: The justification for using antibiotic concentrations for VGS impregnation should be provided in the Materials and Methods section, rather than in the Results section.
Lines 210-211: “Fosfomycin concentrations were selected based on previous studies where rifampicin was impregnated in similar grafts [3].”
Please provide in the M&M section more detailed information about the reasons why these specific concentrations of fosfomycin were used to treat the vascular grafts. Has the MIC of fosfomycin for the bacterial strain tested been taken into consideration?
Lines 211-213: “Oritavancin concentrations were selected based on the MIC results. For each strain antibiotic concentrations were adjusted according to the efficiency obtained.”
Again in the M&M section please explain how the concentrations of oritavancin used for impregnating vascular grafts relate to the MICs of the antibiotic?
Figure 1 and the related text. Where are the results obtained using an oritavancin concentration of 64 mg/L?
Lines 227-228, Figure 1 legend: “CFU counts of Staphylococcus epidermidis incubation media at days 1,2,3 and 7 of fosfomycin (F)/oritavancin (O) impregnated VG and control (Graft prepared in Water).”
Please rephrase it to make it more understandable. A possible example is:
The CFU counts of S. epidermidis measured after incubation in media with fosfomycin (F) and/or oritavancin (O) impregnated VG and control (graft prepared in water) at days 1,2,3 and 7.
Please fix all the legends in this manner.
Figure 3 and the related text: The concentration of oritavancin at 64 mg/L is not mentioned in the text. It seems that this is an insufficient concentration and therefore it is not reported. However, if this was included in the study, then the results should be provided.
Please provide with the vancomycin MICs for SE and MRSA strains.
Comments on the Quality of English LanguageThere are some sentences in the text that could be improved, such as the figure legends. I have provided some suggestions for how to improve them in the comments.
Author Response
In general, I have an average opinion of the article. Apparently, they were inspired by their previous work [15], where a synergistic effect was detected when fosfomycin was combined with oritavancin. Therefore, they decided to conduct this study. The authors have examined how the combination of fosfomycin and oritavancin can suppress the growth of gram-positive bacteria on the surface of vascular grafts.
I have some comments that I believe should be considered.
Line 96: Bacterial strains... (not one strain).
strain was corrected to strains
The MIC data for individual drugs in Table 3 does not match the values specified in Table 2.
The slightly different results for MICs of the antibiotics alone in table 2 and 3 are due to specificities of the different protocols regarding antibiotics dilutions:
- In the MICs of the antibiotics alone, a serial dilution of the antibiotics was made in water and supplemented with TweenTM 80 or G6P, when appropriate, and 10 uL of the antibiotic’s solution (10x concentrated) was added to 90 uL of the inoculum.
- In the synergy tests, the antibiotics were directly diluted in medium supplemented with TweenTM 80 or G6P, when appropriate.
This is the only difference in both procedures, suggesting different solubilities of the antibiotics in water and medium.
Additionally, it is unclear which MIC values were used for the FIC calculations, as the MIC values are given in a range. Please provide an explanation.
The MIC used in the FICI calculations were the values obtained in the synergy tests. The range observed in the MICs is a result of 3 independent assays. For clarification we replaced the MIC range for the exact values used for FICI calculations (table 3).
Lines 210-213: The justification for using antibiotic concentrations for VGS impregnation should be provided in the Materials and Methods section, rather than in the Results section. Lines 210-211: “Fosfomycin concentrations were selected based on previous studies where rifampicin was impregnated in similar grafts [3].”
The justification was moved to the Materials and Methods section.
Please provide in the M&M section more detailed information about the reasons why these specific concentrations of fosfomycin were used to treat the vascular grafts. Has the MIC of fosfomycin for the bacterial strain tested been taken into consideration?
The concentrations of antibiotics used for impregnation in the vascular graft cannot be directly correlated to MIC. In fact, in the case of Fosfomycin, values based on MICs were tested. However, such low amounts were not effective, therefore, values in the range of a previous publication which used rifampicin were tested (40mg/mL), and effective concentrations were found.
Lines 211-213: “Oritavancin concentrations were selected based on the MIC results. For each strain antibiotic concentrations were adjusted according to the efficiency obtained.”Again in the M&M section please explain how the concentrations of oritavancin used for impregnating vascular grafts relate to the MICs of the antibiotic? “Oritavancin concentrations were selected based on the MIC results. For each strain antibiotic concentrations were adjusted according to the efficiency obtained.”
The justification was moved to the Materials and Methods section.
The Oritavancin concentrations used to impregnate the vascular grafts are in the ug/mL range, much lower than for fosfomycin. This could be related to the much lower MICs of Oritavancin for the 3 strains tested.
Figure 1 and the related text. Where are the results obtained using an oritavancin concentration of 64 mg/L?
In this figure we do not show oritavancin concentration at 64ug/mL as eradication was obtained with 16 and 32ug/ml in combination with Fosfomycin at 1.25mg/mL.
Lines 227-228, Figure 1 legend: “CFU counts of Staphylococcus epidermidis incubation media at days 1,2,3 and 7 of fosfomycin (F)/oritavancin (O) impregnated VG and control (Graft prepared in Water).”Please rephrase it to make it more understandable. A possible example is: The CFU counts of S. epidermidis measured after incubation in media with fosfomycin (F) and/or oritavancin (O) impregnated VG and control (graft prepared in water) at days 1,2,3 and 7. Please fix all the legends in this manner.
The sentence was rephrased according to the suggestion of the reviewer, in all the legends.
Figure 3 and the related text: The concentration of oritavancin at 64 mg/L is not mentioned in the text. It seems that this is an insufficient concentration and therefore it is not reported. However, if this was included in the study, then the results should be provided.
Thank you for the suggestion, however for this specific strain, oritavancin at 64ug/mL alone was not bactericidal as well as 128ug/ml. We removed the results for 64ug/mL (oritavancin) combined with 40mg/mL (Fosfomycin) from Figure 3, since even at 128ug/mL (oritavancin) combined with fosfomycin (40mg/mL) was not bactericidal.
Please provide with the vancomycin MICs for SE and MRSA strains.
We believe this information is not relevant for the present study and could introduce confusion on the analysis.
There are some sentences in the text that could be improved, such as the figure legends. I have provided some suggestions for how to improve them in the comments.
Improvements were made in the text of figure legends.

Round 2
Reviewer 2 Report
Comments and Suggestions for Authors
1. The authors wrote:
>>we believe that the test we performed is more suitable for the application of the grafts
So, I'm not believe and it will be more informative to see the results of standardized test.
2. Sorry for my mistake on oritavancin, but the main question remains unanswered: What about cytotoxicity of the prepared materials sample? There is no any evaluation of possible toxic effect.
Author Response
Comment 1:
The authors wrote:
>>we believe that the test we performed is more suitable for the application of the grafts
So, I'm not believe and it will be more informative to see the results of standardized test.
Response 1:
We understand your concern. However, as we previously explained, the agar diffusion test has several limitations that were addressed by the test performed in this study. As mentioned in our previous responde, our method allowed for precise quantification of the bacterial burden eliminated by the grafts while simulating graft-liquid (blood) contact.
Additionally, we employed the methodology used by X. Berard et al. [3] to ensure comparability with this relevant study in the field, where commercial grafts were impregnated with other antibiotics.
Comment 2:
Sorry for my mistake on oritavancin, but the main question remains unanswered: What about cytotoxicity of the prepared materials sample? There is no any evaluation of possible toxic effect.
Response 2:
The primary aim of this study was to assess the efficacy of the combined impregnation of fosfomycin and oritavancin in eliminating relevant bacterial strains in graft-associated infections, including antibiotic-resistant strains. Since both fosfomycin and oritavancin are already in clinical use, their cytotoxicity has been extensively studied. However, we acknowledge that the cytotoxicity of the grafts impregnated with this combination of antibiotics needs to characterized. This will be addressed in future pre-clinical studies.
Round 3
Reviewer 2 Report
Comments and Suggestions for Authors
So, the current version of the manuscript contains the results of just 1 experiment. It seems promising and interesting for further pre-clinical research but in my opinion it's not enough for publication. I recommend to reject the manuscript due to lack of scientific novelty and incomplete character of the work.
Author Response
Response:
Thank you for considering our justifications.
The statements suggested were added to the text and are highlighted in yellow in the manuscript document as follows:
Lines 157 – 161 “This methodology was used as an alternative to the standard agar diffusion plate test (ISO 20645:2004) for determining the antibacterial activity of textile/fiber materials, allowing for precise quantification of the bacterial burden eliminated by the grafts while simulating graft-liquid (blood) contact.”
Lines 371- 375 “Although both antibiotics are already in clinical use and their biocompatibility has been extensively studied [29,30], with no common side effects reported, the potential for allergic reactions should still be considered [31, 32].”